# Strain Rate Changes during Stress Echocardiography Are the Most Accurate Predictors of Significant Coronary Artery Disease in Patients with Previously Treated Acute Coronary Syndrome

**DOI:** 10.3390/diagnostics13101796

**Published:** 2023-05-19

**Authors:** Rafik Shenouda, Ibadete Bytyçi, Eman El Sharkawy, Noha Hisham, Mohamed Sobhy, Michael Y. Henein

**Affiliations:** 1Institute of Public Health and Clinical Medicine, Umea University, 90187 Umea, Sweden; rafik.shenouda@umu.se (R.S.);; 2International Cardiac Centre, Alexandria 21526, Egypt; 3Clinic of Cardiology, University Clinical Centre of Kosovo, 10000 Prishtina, Kosovo; 4Cardiology Department, Faculty of Medicine, Alexandria University, Alexandria 21500, Egypt; 5Molecular and Clinic Research Institute, St. George University, London SW17 0QT, UK

**Keywords:** acute coronary syndrome, dobutamine stress echocardiography, myocardial deformation parameters

## Abstract

Background and Aims. Dobutamine stress echocardiography (DSE) is a well-established non-invasive investigation for the detection of ischemic myocardial dysfunction. The aim of this study was to evaluate the accuracy of myocardial deformation parameters measured by speckle tracking echocardiography (STE) in predicting culprit coronary artery lesions in patients with prior revascularization and acute coronary syndrome (ACS). Methods. We prospectively studied 33 patients with ischemic heart disease, a history of at least one episode of ACS and prior revascularization. All patients underwent a complete stress Doppler echocardiographic examination, including the myocardial deformation parameters of peak systolic strain (PSS), peak systolic strain rate (SR) and wall motion score index (WMSI). The regional PSS and SR were analyzed for different culprit lesions. Results. The mean age of patients was 59 ± 11 years and 72.7% were males. At peak dobutamine stress, the change in regional PSS and SR in territories supplied by the LAD showed smaller increases compared to those in patients without culprit LAD lesions (*p* < 0.05 for all). Likewise, the regional parameters of myocardial deformation were reduced in patients with culprit LCx lesions compared to those with non-culprit LCx lesions and in patients with culprit RCA legions compared to those with non-culprit RCA lesions (*p* < 0.05 for all). In the multivariate analysis, the △ regional PSS (1.134 (CI = 1.059–3.315, *p* = 0.02)) and the △ regional SR (1.566 (CI = 1.191–9.013, *p* = 0.001)) for LAD territories predicted the presence of LAD lesions. Similarly, in a multivariable analysis, the △ regional PSS and the △SR predicted LCx culprit lesions and RCA culprit lesions (*p* < 0.05 for all). In an ROC analysis, the PSS and SR had higher accuracies compared to the regional WMSI in predicting culprit lesions. A △ regional SR of −0.24 for the LAD territories was 88% sensitive and 76% specific (AUC = 0.75; *p* < 0.001), a △ regional PSS of −1.20 was 78% sensitive and 71% specific (AUC = 0.76, *p* < 0.001) and a △ WMSI of −0.35 was 67% sensitive and 68% specific (AUC = 0.68, *p* = 0.02) in predicting LAD culprit lesions. Similarly, the △ SR for LCx and RCA territories had higher accuracies in predicting LCx and RCA culprit lesions. Conclusions. The myocardial deformation parameters, particularly the change in regional strain rate, are the most powerful predictors of culprit lesions. These findings strengthen the role of myocardial deformation in increasing the accuracy of DSE analyses in patients with prior cardiac events and revascularization.

## 1. Introduction

Globally, in 2017, ischemic heart diseases (IHD) affected 126.5 million people worldwide, an increase of 74.9% compared to 1990 [1]. A total of 10% to 15% of patients presenting to the emergency department with symptoms suggestive of acute coronary syndrome (ACS) proved to have the syndrome [2]. Many tools for cardiac investigations have been developed over the years to enhance the accuracy of diagnosing ACS and its risk stratification [3], including ECG, conventional echocardiography, radioisotope myocardial perfusion scanning, computed tomography coronary angiography and cardiac magnetic resonance imaging. One of these rapidly developing tools is echocardiography, as it is the most patient friendly technique and can give comprehensive information about different aspects of the function of the cardiac muscle in a relatively short examination time, with its different modalities: two-dimensional, M-mode, myocardial doppler velocities and myocardial deformation parameters. These different modalities could be used at rest and during stress, whether that is pharmacological stress or physical exercise. Deformation parameter measurements in the form of myocardial strain and strain rate have proven to be accurate in diagnosing coronary artery disease [4,5,6,7,8]. Dobutamine, dipyridamole and adenosine are among the pharmacological drugs used to induce myocardial stress by different mechanisms, either by increasing the myocardial oxygen demand through increasing the heart rate with dobutamine or by inducing coronary vasodilatation with dipyridamole and adenosine. The latter two drugs are commonly used during radioisotope cardiac investigations [9]. Dobutamine stress echocardiography (DSE) has demonstrated a substantial clinical relevance in the detection of ischemic myocardial dysfunction based on its high sensitivity and specificity [10,11]. Stress-induced ischemia results in the development of new or worsening regional wall motion abnormalities in the region subtended by the significantly stenosed coronary artery [12]. In addition to its diagnostic value, stress echocardiography is a very effective prognostic tool in chronic coronary artery disease, after myocardial infarction, and in the evaluation of patients prior to major non-cardiac surgery [13,14]. Furthermore, it has proven to be very accurate in predicting the functional recovery of segmental wall motion abnormalities after revascularization, hence providing valuable physiological information for patients being considered for valve surgery [15,16].

Visual assessment of segmental wall motion in the form of wall thickening and inward endocardial movement remains the primary method of analysis of stress echocardiograms [17,18]. However, an objective assessment of myocardial deformation in the form of strain and strain rate could be more accurate than the subjective visual assessment [19,20]. In addition to myocardial ischemia, detection of viable myocardium is a crucial indicator in stress echocardiography, either prior to coronary revascularization or post angiography in order to plan a sound revascularization strategy [21]. Exercise echocardiography is performed using physical exercise on a treadmill, bicycle or semi-supine bicycle. Such a versatile application makes stress echocardiography an essential cardiac investigation in daily practice, not only in cardiac centers but also in district general hospitals.

The aim of this study was to compare the accuracy of two-dimensional wall motion assessments of ischemia in patients with chronic stable angina during conventional dobutamine stress echocardiography, with myocardial deformation parameters measured by speckle tracking echocardiography (STE), in predicting the culprit coronary artery lesions in patients with prior revascularization and ACS.

## 2. Methods

### 2.1. Study Protocol

Thirty-three patients were recruited in this study who presented to the International Cardiac Center hospital (ICC) in Alexandria, Egypt, between September 2021 and October 2022 because of symptoms suggestive of ischemia (mainly chest pain, breathlessness or both). All patients had a previous history of at least one episode of acute coronary syndrome, percutaneous coronary intervention (PCI) or coronary artery bypass graft (CABG) surgery.

All patients underwent DSE to assess the presence and/or extent of ischemia to justify a referral for a coronary angiogram. Written consent was obtained from each patient and a full medical history was taken by a cardiology specialist, with details about coronary artery disease risk factors including hypertension, diabetes mellitus, dyslipidemia, smoking history, obesity and family history of CAD.

Echocardiographic examination: An intravenous line was established in all patients and ECG lead II was connected. A baseline echocardiogram using an Affinity 30 echocardiograph (Philips corporation, Cambridge, MA, USA) was performed on all patients with visually assessed wall motion abnormalities, applying the conventional segmental scoring system: (1) normal or hyperkinetic, (2) hypokinetic (reduced thickening or motion), (3) akinetic (absent or negligible thickening) and (4) dyskinetic (systolic thinning or stretching or aneurysm). The left ventricular ejection fraction (EF) was estimated using the modified Simpson method or conventional M-mode measurements. The presence of mitral regurgitation (if any) was noted, and the severity was assessed according to published guidelines [22]. Dobutamine was prepared and given via the intravenous line using a commercial syringe pump with a starting and an incremental dose of 5 mic/kg/min every 3 min until a maximum dose of 40 mic/kg/min. Wall motion changes, the ejection fraction (EF) and mitral regurgitation were all evaluated at the end of each stress stage. The test was terminated if the maximum dose of 40 µg/kg/min of dobutamine was achieved; the patient developed symptoms of chest pain or shortness of breath; the patient developed clear signs of ischemic ECG changes; or a 2 mm ST shift, arrhythmia, couplets, runs of ectopic beats, atrial fibrillation or a systolic blood pressure drop of 20 mm Hg were detected. Oxygen saturation was also continuously monitored and recorded at rest and at the end of each stress stage [23].

After the test, an offline analysis of myocardial deformation parameters including peak systolic strain (PSS) and peak systolic strain rate (SR) was made using speckle tracking echocardiography (STE) technology software (Q lab 15 Philips, USA). Resting and peak stress images and measurements were compared. Speckle tracking images were obtained with patients in the left lateral decubitus position from apical 4-, 3- and 2-chamber views at rest and at peak stress. Two-dimensional greyscale images were obtained at a frame rate of 70–80 Hz during three cardiac cycles and were then digitally stored for offline analysis. An offline analysis of the STE cine loops was performed using commercially available software (Q lab 15 Philips, Cambridge, MA, USA). After manually outlining the clear myocardial border at the end systole, the ROI (the region of interest), generated automatically by the software, was manually adjusted and the peak systolic strain (PSS) and the peak systolic strain rate (SR) at basal, middle and apical levels were all determined [22,24].

Coronary angiogram: DSE examination was performed prior to coronary angiography and possible revascularization was performed based on the results. All patients underwent a coronary angiogram using the Judkins procedure and an onsite Toshiba or Philips lab using the conventional protocol.

Inclusion criteria: Patients who presented with new onset of dyspnea, chest pains or both and were free of symptoms after their last coronary episode with a history of at least one episode of ACS and patients with reduced resting EF by conventional echocardiography or available data of a recent coronary.

Exclusion criteria: Patients with a permanent pacemaker (PPM), an intra cardiac defibrillator (ICD) or cardiac resynchronization therapy devices (CRT) and patients with left ventricular (LV) thrombus or more than mild valve regurgitation. Additionally, we excluded patients with overt signs of heart failure, patients with a high NT-pro BNP and patients with other causes of dyspnea such as chest infections or anemia.

This study conformed with the Helsinki convention guidelines, the protocol was approved by the local Ethics Committee (approval number: 6/2021) and all patients gave informed consent to participate in the study.

### 2.2. Statistical Analysis

Statistical analyses were performed using SPSS Software Package version 26.0 (IBM Corp., Armonk, NY, USA). Values are expressed as means ± standard deviation (SD) for continuous variables and frequencies (percentage) for categorical variables. Continuous data were compared with a two-tailed Student’s *t* test and discrete data were compared with a Chi-square test. Variables were compared between any two groups using an unpaired Student’s *t* test or a Fisher’s exact probability test. The degrees of association between clinical and biochemical scales were determined using Pearson’s correlation coefficient in the case of continuous variables, a Chi-square test for categorical variables and point biserial correlation for categorical and continuous variables.

Predictors of the culprit lesions were identified using a univariate analysis and multivariate logistic regression. The method and the number of predictors in the model, the anticipated effect size and the desired statistical power level were calculated. As a result of the small sample size, to have a desired statistical power level of about 0.8, we calculated the a priori sample size for multiple regression. Receiver operational characteristic (ROC) analyses were performed, and the best cut-off value, sensitivity and specificity were determined. A significant difference was defined as a *p* value of <0.05 (two-tailed).

## 3. Results

### Demographic and Clinical Indices of the Patients

Of the 33 studied patients with a mean age of 59 ± 11 years, 72.7% were males. A total of 17 (51.5%) had diabetes, 16 (48.5%) had dyslipidemia, 18 (54%) had hypertension, 42.4% smoked and 7 (21.2%) were obese. Eighteen patients had a previous PCI (54.5%) and nine patients had a previous CABG (27.3%). Out of 33 patients, 18 patients (54.5%) presented with typical chest pain, 9 (27.2%) patients presented with breathlessness and 6 patients (18.2%) had mixed symptoms (Table 1).

## 4. Coronary Angiography Findings

Significant luminal stenosis was found in 16 patients with left anterior descending (LAD) artery lesions, in 25 patients with left circumflex (LCx) artery lesions and in 25 patients with right coronary artery (RCA) lesions, while sub occlusion was found in 17 patients with LAD lesions, 8 patients with LCx artery lesions and 8 patients with RCA lesions (Table 1).

### 4.1. Echocardiographic Indices

At rest, the mean LV EF was 36.2 ± 8.9%, the global wall motion score index was 2.07 ± 0.5 and the parameters of global myocardial deformation (PSS and SR) were 10.4 ± 2.6 and −0.55 ± 0.2, respectively, without significant difference in sub-regions supplied by different arteries (*p* > 0.05 for all; Appendix A).

At peak stress, the LV global systolic function including LVEF and PSS showed similar delta changes (−6.03 ± 4.2 (16.6%) vs. −1.81 ± 0.9 (17.3%)), while the delta SR was −0.23 ± 0.6 (41.8%). Likewise, the delta global WMSI was 0.22 ± 0.2 (10.6%), with a similar extent in territories supplied by LAD and LCx (0.27 ± 0.23 (12.0%) vs. 0.20 ± 0.4 (11.8%)) but significantly less (0.11 ± 0.3 (4.84%)) in the territories supplied by RCA (Table 2). In the subgroup analysis, patients with culprit LAD lesions showed a systolic global LV function increase compared to resting values (*p* < 0.0) but to a lesser extent compared to patients without culprit LAD (LVEF (8.0 ± 4.3 (21.0%) vs. 3.2 ± 3.5 (10.6%); *p* = 0.005), PSS (−2.9 ± 0.9 (25.4%) vs. −0.9 ± 1.1 (9.9%); *p* < 0.01), SR (−0.29 ± 1.1 (48.3%) vs. −0.17 ± 0.2 (34.1%); *p* = 0.02)). Similarly, the longitudinal PSS and SR in the territories supplied by LAD showed a lower increase in patients with culprit LAD lesions compared to those without culprit LAD lesions (*p* < 0.05 for all). The other territories supplied by RCA and LCx did not show any difference between the two respective groups. WMSI divided by territories did not show any significant difference (Table 3 and Appendix A). Likewise, the global and regional parameters of myocardial deformation were worse in patients with culprit LCx compared to non-culprit LCx and in patients with culprit RCA compared to those with non-culprit RCA lesions (*p* < 0.05 for all, Table 3).

### 4.2. Correlation of CAD Significance with Global and Regional LV Systolic Function

A modest correlation was found between △ LVEF (r = 0.50, *p* = 0.001), △ global SR (r = 0.52, *p* = 0 < 0.001), △ global PSS (r = 0.43, *p* = 0.02) and △ global WMSI (r = 0.40, *p* = 0.02) in patients with significant LAD lesions, while △ SR in territories supplied by the LAD had a higher correlation (r = 0.60, *p* < 0.001). Similarly, a significant correlation was found between △ LVEF (r = 0.51, *p* = 0.01), △ global SR (r = 0.49, *p* = 0.01), △ global PSS (r = 0.46, *p* = 0.02) and △ global WMSI (r = 0.37, *p* = 0.04) in patients with significant LCx lesions. Additionally, the △ SR in territories supplied by the LCx had moderate correlation with significant LAD lesions (r = 0.55; *p* < 0.001). Likewise, there was a modest correlation between the global systolic function parameters and RCA significant lesions (Figure 1A–F and Appendix A).

### 4.3. Echocardiographic Predictors of Culprit Lesions

In the univariate analysis, △ regional PSS of segments supplied by the LAD (*p* = 0.02) and △ regional SR of the same LAD segments (*p* = 0.001) predicted culprit LAD lesions. △ regional PSS and △ regional SR also were the univariate predictors of RCA and LCx culprit arteries (*p* < 0.05 for all), but the regional WMSI significantly predicted the culprit lesion in LCx disease.

In the multivariate analysis, △ regional PSS (1.134 (CI = 1.059–3.315, *p* = 0.02)) and △ regional SR (1.566 (CI = 1.191–9.013, *p* = 0.001)) for territories supplied by the LAD predicted LAD lesions. Similarly, in the multivariable analysis, △ regional PSS and △ SR were also predictors of LCx culprit lesions and RCA culprit lesions (*p* < 0.05 for all, Appendix A).

Using an ROC analysis, the regional myocardial deformation, including PSS and SR, had a higher accuracy compared to the regional WMSI in predicting culprit lesions. A △ SR of −0.24 for the LAD territories was 88% sensitive and 76% specific (AUC = 0.75; *p* < 0.001), a △ PSS of −1.20 for LAD territories was 78% sensitive and 71% specific (AUC = 0.76, *p* < 0.001) and a △ WMSI of −0.35 for LAD territories was 67% sensitive and 68% specific (AUC = 0.68, *p* = 0.02) in predicting LAD culprit lesions (Figure 2A). Similarly, a △ SR of −0.24 for LCx and RCA territories had a higher accuracy in predicting LCx and RCA culprit lesions (Figure 2B,C).

## 5. Discussion

Findings: The results of this paper can be summarized as follows: (1) in patients with prior ACS and revascularization, the myocardial deformation parameters are more accurate than the conventional wall motion score index in detecting the coronary lesions responsible for recent admission/symptom development; (2) the myocardial longitudinal SR was consistently more accurate than the myocardial PSS in predicting the culprit lesions; (3) the myocardial SR accuracy in predicting the culprit lesions was higher in the territories subtended by the LAD and LCx arteries compared to those subtended by the right coronary arteries.

Data interpretation: Although the wall motion score index is currently used as the conventional echocardiographic parameter for diagnosing ischemic dysfunction [25] and for diagnosing significant coronary artery stenosis (>70%) according to guidelines [26], it proved less accurate in the subset of patients we studied compared to myocardial deformation measurements. All our patients had symptoms suggestive of ischemia, probably stable angina, but we used stress echocardiography on them in order to identify the most accurate echocardiographic parameter for predicting the culprit lesion and whether they are the same as we previously found in patients with acute coronary syndrome (STEMI and NSTEMI) [27,28]. This objective was also strengthened by the underlying ischemic myocardial disease our patients had, as manifested by a reduced resting LV EF. Indeed, our results showed the myocardial SR as the most accurate parameter compared with the myocardial PSS and the conventional WMSI in detecting culprit lesions. The reason behind such a difference in accuracy is three-fold. Firstly, myocardial deformation reflects the intrinsic longitudinal function, which itself reflects the function of the sub-endocardium, the most sensitive myocardial layer to ischemia [29,30]. This is in contrast to a wall motion analysis, which reflects the transvers full thickness myocardial circumferential function. Secondly, the myocardial SR represents the velocity of deformation and shortening of the longitudinal segments with respect to the original segmental length. This finding supports our previously reported findings before and during coronary angioplasty, where myocardial segmental velocities proved the most sensitive in reflecting acute ischemia [31,32] as well as successful revascularization [30,33]. Thirdly, it seems that the accuracy of SR changes in predicting culprit lesions is higher for the left coronary system compared to the right, in particular the LAD subtended segments. This finding is difficult to interpret because of the variability of the dominant system, which could be the circumflex or right coronary artery. Finally, our findings are compatible with what we have previously reported in patients with STEMI and NSTEMI [34,35], thus strengthening the accuracy of the myocardial SR in predicting ischemia and the impact of coronary disease, irrespective of the clinical presentation, whether at first admission or after prior interventions.

The WMSI reflects the extent of inward displacement of different myocardial segments around the LV cavity, which is of course also contributed to by segmental systolic thickening. Such a function is controlled mainly by the circumferential myocardial muscle fibers of the ventricle. On the other hand, the longitudinal deformation function is mainly contributed to by shortening and lengthening of the subendocardial myocardial fibers, which are the most sensitive layer to ischemia. Such anatomical differences explain the different accuracies between the two methods we compared. Finally, even comparing the two longitudinal myocardial deformation parameters showed the superior accuracy of the strain rate in predicting culprit lesions over and above that of the myocardial strain. This finding is compatible with what we previously documented as a result of successful coronary angioplasty, despite using a different modality [36,37]. Thus, our results suggest a faster myocardial velocity response to ischemia well before segmental amplitude changes become obvious.

Clinical implications: In patients with complex coronary artery disease and prior cardiac events, myocardial deformation in the form of the SR is the most accurate marker of ischemia and the strongest predictor of coronary culprit lesions that need revascularization. Although our findings are not exactly in line with the current guidelines, they could be seen as eye opener for the additional application of stress echocardiography in symptomatic patients irrespective of the acuity of their condition or the occurrence of prior events, particularly for those with no additional significant valve diseases.

Study Limitations: Our study has limitations. We could not perform power calculations because of the lack of previous publications for similar groups of patients. This limitation resulted in underpowering the study. This is shown clearly when dividing the patients into subgroups. We would have liked to study patients with prior myocardial infarctions and compare them with those with unstable angina, but the small numbers limited the relevance of this objective. Additionally, we would have liked to compare myocardial deformation parameters with other echo modalities in the same group of patients, including tissue Doppler velocities, but the need for a fast acquisition time prohibited achieving this desire.

## 6. Conclusions

Myocardial deformation in the form of the SR is a powerful predictor of culprit coronary artery lesions in symptomatic patients with prior coronary artery interventions. These findings add to the valuable application of myocardial strain rate measurements as part of stress echocardiography, irrespective of patient presentation.

## Figures and Tables

**Figure 1 diagnostics-13-01796-f001:**
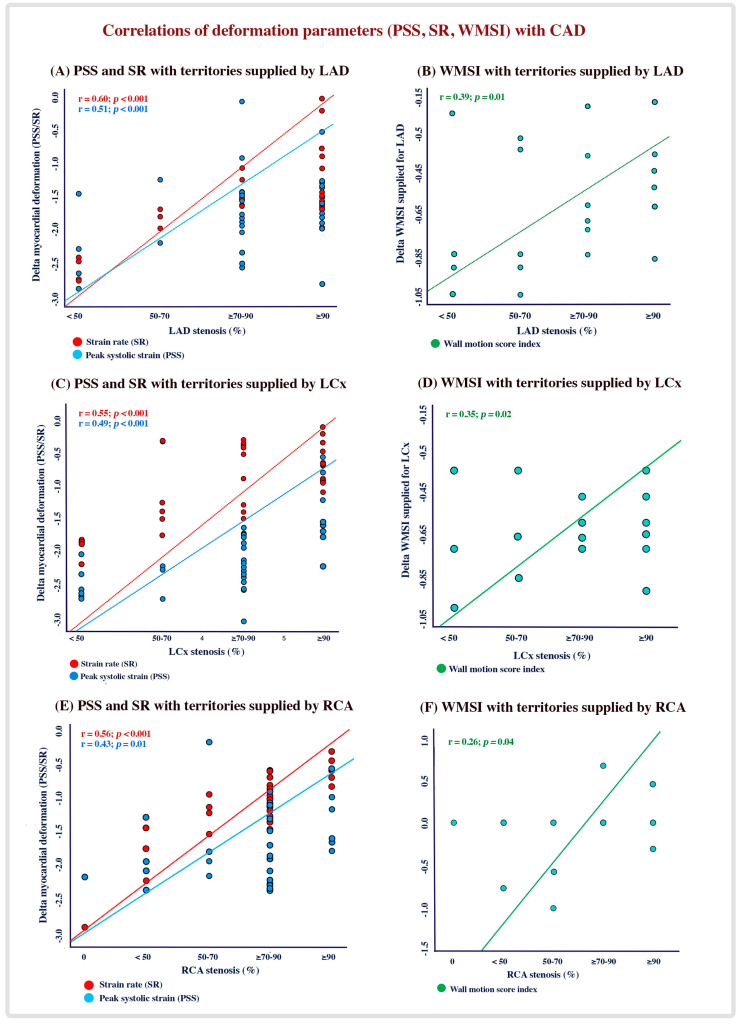
Correlation between deformation parameters and WMSI with the degree of CAD.

**Figure 2 diagnostics-13-01796-f002:**
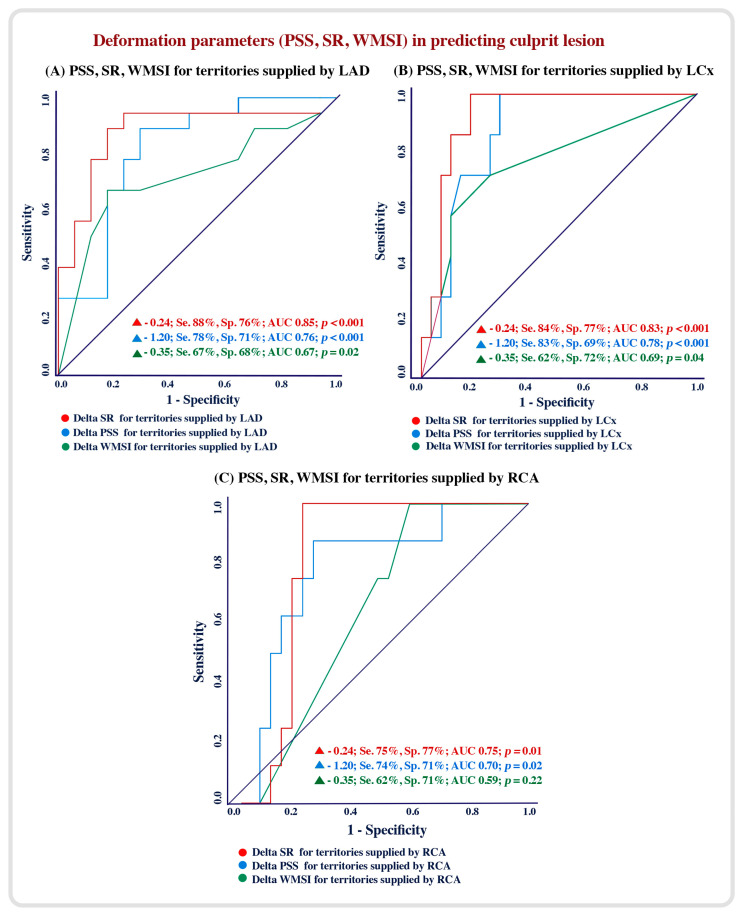
Accuracy of deformation parameters and WMSI in predicting culprit lesions.

**Table 1 diagnostics-13-01796-t001:** Patient demographics and clinical indices.

Variable	Patients
	(n = 33)
Demographics and clinical indices	
Age (years)	59 ± 11
Males (n, %)	24 (72.7)
Hypertension (n, %)	18 (54.5)
Diabetes (n, %)	17 (51.5)
Dyslipidemia (n, %)	16 (48.5)
Obesity (n, %)	7 (21.2)
Smokers (n, %)	14 (42.4)
Previous PCI (n, %)	18 (54.5)
Previous CABG (n, %)	9 (27.3)
Clinical presentation	
Chest pain (n, %)	18 (54.5)
Dyspnea (n, %)	9 (27.2)
Mixed symptoms (n, %)	6 (18.2)
Outcome data	
Time since first ACS (years)	6.10 ± 3.9
Culprit (occluded) lesion (n, %)	23 (69.7)
LAD occluded (n, %)	17 (51.5)
LCx occluded (n, %)	8 (24.2)
RCA occluded (n, %)	8 (24.2)

PCI: percutaneous coronary intervention; CABG: coronary artery bypass grafting. LAD: left anterior descending; LCx: left circumflex; RCA: right coronary artery.

**Table 2 diagnostics-13-01796-t002:** Delta changes (baseline and peak stress) of echocardiographic parameters.

Variable		All Patients
		(n = 33)
LV systolic function		
LV EF (%)	Δ (%)	−6.03 ± 4.2 (16.6)
Global PSS	Δ (%)	−1.81 ± 0.9 (17.3)
Global SR	Δ (%)	−0.23 ± 0.6 (41.8)
Territories supplied by LAD		
Longitudinal PSS	Δ (%)	−1.37 ± 1.5 (13.9)
Strain rate	Δ (%)	−0.13 ± 0.4 (23.2)
Territories supplied by LCx		
Longitudinal PSS	Δ (%)	−1.83 ± 1.1 (16.9)
Strain rate	Δ (%)	−0.28 ± 0.2 (54.7)
Territories supplied by RCA		
Longitudinal PSS	Δ (%)	−1.65 ± 1.2 (15.4)
Strain rate	Δ (%)	−0.24 ± 0.4 (44.4)
Wall motion score index		
Global WMSI	Δ (%)	0.22 ± 0.2 (10.6)
WMSI supplied by LAD	Δ (%)	0.27 ± 0.3 (12.0)
WMSI supplied by LCx	Δ (%)	0.20 ± 0.4 (11.8)
WMSI supplied by RCA	Δ (%)	0.11 ± 0.3 (4.84)

Abbreviations: EF, ejection fraction; PSS, peak systolic strain; LAD, left anterior descending artery; LCx, left circumflex artery; RCA, right coronary artery; WMSI, wall motion score index. Note: territories supplied by LAD (basal anterior septum, mid-anterior septum, apical septum, basal-anterior, mid-anterior, apical anterior and apex); territories supplied by LCx (basal antro-lateral, mid antro-lateral, basal-infero-lateral, mid-infero-lateral and apico-lateral); territories supplied by RCA (basal-inferior, mid-inferior, apical-inferior, basal infero-septal and mid-infero-septal).

**Table 3 diagnostics-13-01796-t003:** Delta changes of echocardiographic indices in patient subgroups.

Variable		Patients	Patients	Patients	Patients	Patients	Patients
		LAD −	LAD +	LCx −	LCx +	RCA −	RCA +
		(n = 16)	(n = 17)	(n = 25)	(n = 8)	(n = 25)	(n = 8)
LV systolic function							
LV EF (%)	Δ (%)	8.9 ± 4.3 (21.0)	3.2 ± 3.5 (10.6)	6.7 ± 4.3 (17.3)	4.1 ± 3.3 (13.6)	8.5 ± 4.6 (20.7)	5.9 ± 4.2 (18.7)
Global PSS	Δ (%)	−2.9 ± 0.9 (25.4)	−0.9 ± 1.1 (9.9)	−2.1 ± 1.2 (19.2)	−0.9 ± 1.1 (9.8)	−2.30 ± 0.9 (19.8)	−1.10 ± 1.2 (11.1)
Global SR	Δ (%)	−0.29 ± 1.1 (48.3)	−0.17 ± 0.2 (34.1)	−0.33 ± 0.2 (55.9)	−0.14 ± 0.2 (37.8)	−0.29 ± 0.2 (42.6)	−0.14 ± 0.2 (27.4)
Territories supplied by LAD							
Longitudinal PSS	Δ (%)	−1.73 ± 0.9 (15.4)	−1.0 ± 0.9 (11.8)	−1.62 ± 1.4 (16.2)	−0.86 ± 0.9 (9.42)	−1.9 ± 1.4 (17.9)	−1.17 ± 1.6 (12.1)
Strain rate	Δ (%)	−0.17 ± 0.2 (28.3)	−0.02 ± 0.3 (3.9)	0.14 ± 0.3 (25.4)	0.14 ± 0.3 (22.4)	−0.06 ± 0.4 (10.3)	−0.08 ± 0.2 (13.1)
Territories supplied by LCx							
Longitudinal PSS	Δ (%)	−1.89 ± 1.1 (16.3)	−1.77 ± 1.1 (17.9)	−2.3 ± 1.1 (19.9)	−0.7 ± 0.3 (8.2)	−1.80 ± 1.0 (14.7)	−1.70 ± 1.1 (15.6)
Strain rate	Δ (%)	−0.33 ± 0.3 (61.1)	−0.29 ± 0.3 (60.3)	−0.31 ± 0.3 (50.1)	−0.10 ± 0.4 (24.3)	−0.31 ± 1.6 (58.4)	−0.27 ± 0.9 (43.5)
Territories supplied by RCA							
Longitudinal PSS	Δ (%)	−0.31 ± 0.3 (20.8)	−1.2 ± 1.1 (12.2)	−1.21 ± 1.7 (9.52)	−1.20 ± 1.6 (10.4)	2.70 ± 1.6 (22.3)	−0.66 ± 0.9 (6.8)
Strain rate	Δ (%)	−0.24 ± 0.1 (42.1)	−0.24 ± 0.2 (46.1)	−0.13 ± 0.2 (18.1)	−0.08 ± 0.2 (11.3)	−0.26 ± 0.5 (37.1)	−0.05 ± 0.2 (8.9)
Wall motion score index							
Global WMSI	Δ (%)	0.43 ± 0.2 (33.5)	0.16 ± 0.2 (8.18)	0.27 ± 0.2 (13.4)	0.21 ± 0.2 (9.21)	0.27 ± 0.2 (13.1)	0.40 ± 0.2 (18.1)
WMSI supplied by LAD	Δ (%)	0.32 ± 0.2 (17.4)	0.22 ± 0.3 (8.43)	0.24 ± 0.3 (10.7)	0.27 ± 0.4 (12.1)	0.28 ± 0.2 (12.4)	0.21 ± 0.3 (9.9)
WMSI supplied by LCx	Δ (%)	0.44 ± 0.3 (26.6)	0.43 ± 0.4 (24.5)	0.37 ± 0.3 (22.8)	0.52 ± 0.3 (26.6)	0.38 ± 0.4 (29.6)	0.66 ± 0.5 (56.4)
WMSI supplied by RCA	Δ (%)	0.06 ± 0.3 (2.69)	0.09 ± 0.2 (3.89)	0.08 ± 0.3 (3.70)	0.04 ± 0.2 (1.50)	0.02 ± 0.2 (0.93)	0.37 ± 0.5 (13.7)

Abbreviations: (−) significant but not occluded; (+) occluded; EF, ejection fraction; PSS, peak systolic strain; LAD, left anterior descending artery; LCx, left circumflex artery; RCA, right coronary artery; WMSI, wall motion score index.

## Data Availability

The authors confirm that the data supporting the findings of this study are available within the article and its Appendix A.

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
