# Peer review of "Strain Rate Changes during Stress Echocardiography Are the Most Accurate Predictors of Significant Coronary Artery Disease in Patients with Previously Treated Acute Coronary Syndrome"

_diagnostics, 2023, doi:10.3390/diagnostics13101796_

Round 1

Reviewer 1 Report

Dear authors,

I want to thank you for submitting your publication on the use of speckle tracking echocardiography in identifying coronary artery disease. Your study gives useful insights into the potential of STE as a diagnostic tool and has the potential to change clinical practise in a substantial manner.

Overall, I thought your study to be well-designed and well-executed, with clear research objectives and acceptable methodology. Nonetheless, there are several sections where I feel your book may be enhanced.

Firstly, in the introductory part, I propose giving greater information on the frequency and significance of acute coronary syndrome and the current diagnostic tools available for diagnosing coronary artery disease. Moreover, a better definition of the study topic and hypothesis would be good.

Secondly, in the methods section, I propose giving greater explanation on the inclusion and exclusion criteria for the research participants, as well as the particular methodologies employed for speckle tracking echocardiography analysis. Additional information on how the areas of interest were picked and how the strain and strain rate measurements were generated would also be useful.

Lastly, in the results section, I propose offering a better description of the major findings and emphasising the most relevant outcomes. Moreover, presenting more thorough statistical analyses and giving confidence ranges for the observed impact sizes would be advantageous.

Fourthly, in the discussion section, I propose rewriting to offer a more extensive analysis of the data in the light of past studies on the utility of STE in identifying coronary artery disease. You might also highlight the possible therapeutic consequences of your results and recommend future research approaches.

Lastly, in the language and writing style, I propose editing the document to increase the clarity and readability of the language. This might involve utilising simpler language patterns, minimising jargon and technical vocabulary where feasible, and maintaining uniform formatting and terminology across the work.

Additionally, here are some recommended modifications in this article:

1. Give further background information on the relevance of myocardial deformation parameters and how they may be utilised to diagnose coronary artery disease using 5 to 7 current studies.

2. Clearly outline and explain the technique utilised in the research, including the selection criteria for participants and the statistical analysis done.

3. Give additional precise information on the imaging modalities employed in the research, especially the details of the stress Doppler echocardiographic assessment.

4. Consider using a graphic or figure to visually convey the outcomes of the investigation.

5. Employ more explicit terminology when summarising the outcomes of the research, and avoid using ambiguous phrases such as "worse".

6. Give more extensive information on the limits of the research, including any possible confounding variables or sources of bias.

7. Consider incorporating a discussion section that gives a more in-depth examination and interpretation of the data.

8. Use more current references to corroborate the study's conclusions, since the references given in the paper are fairly old.

9. Consider adding a remark about the clinical significance of the study's results and how they may be applied in practise.

10. Give a clear and succinct description of the major results and implications of the research at the beginning of the article, in order to offer readers with an initial comprehension of the study's relevance.

Overall, your work has the potential to make a substantial addition to the field of cardiology, and I advise you to address the areas for improvement to enhance your paper. Thank you again for your submission, and I look forward to reviewing the corrected paper.

In the language and writing style, I propose editing the document to increase the clarity and readability of the language. This might involve utilising simpler language patterns, minimising jargon and technical vocabulary where feasible, and maintaining uniform formatting and terminology across the work for example the term "Worse".

Author Response

Reviewer #1

Comments and Suggestions for Authors

Dear authors,

I want to thank you for submitting your publication on the use of speckle tracking echocardiography in identifying coronary artery disease. Your study gives useful insights into the potential of STE as a diagnostic tool and has the potential to change clinical practice in a substantial manner.

Overall, I thought your study to be well-designed and well-executed, with clear research objectives and acceptable methodology. Nonetheless, there are several sections where I feel your book may be enhanced.

Firstly, in the introductory part, I propose giving greater information on the frequency and significance of acute coronary syndrome and the current diagnostic tools available for diagnosing coronary artery disease. Moreover, a better definition of the study topic and hypothesis would be good.

Response: Thank you for your comment. We have corrected this in the revised manuscript.  

Secondly, in the methods section, I propose giving greater explanation on the inclusion and exclusion criteria for the research participants, as well as the particular methodologies employed for speckle tracking echocardiography analysis. Additional information on how the areas of interest were picked and how the strain and strain rate measurements were generated would also be useful.

Response: Thank you for your comment. We have added more details about the inclusion and exclusion criteria and speckle tracking acquirement.

Lastly, in the results section, I propose offering a better description of the major findings and emphasising the most relevant outcomes. Moreover, presenting more thorough statistical analyses and giving confidence ranges for the observed impact sizes would be advantageous.

Response: Thank you for your comment. We have corrected this in the revised manuscript.

Fourthly, in the discussion section, I propose rewriting to offer a more extensive analysis of the data in the light of past studies on the utility of STE in identifying coronary artery disease. You might also highlight the possible therapeutic consequences of your results and recommend future research approaches.

Lastly, in the language and writing style, I propose editing the document to increase the clarity and readability of the language. This might involve utilising simpler language patterns, minimising jargon and technical vocabulary where feasible, and maintaining uniform formatting and terminology across the work.

Response: Thank you for your comment. We have revised the language as recommended.

Additionally, here are some recommended modifications in this article:

  1. Give further background information on the relevance of myocardial deformation parameters and how they may be utilised to diagnose coronary artery disease using 5 to 7 current studies.

Response: Thank you. We have revised the manuscript according to your recommendation.

  1. Clearly outline and explain the technique utilised in the research, including the selection criteria for participants and the statistical analysis done.

Response: Thank you. We have corrected this in the revised manuscript

  1. Give additional precise information on the imaging modalities employed in the research, especially the details of the stress Doppler echocardiographic assessment.

Response: Thank you for your comment. We assessed patients using conventional wall motion abnormalities at rest and peak stress using speckle tracking analysis as well as other conventional echocardiographic modalities. 

Consider using a graphic or figure to visually convey the outcomes of the investigation.

Thank you, we have added a graph as recommended.

  1. Employ more explicit terminology when summarising the outcomes of the research and avoid using ambiguous phrases such as "worse".

 Response: Thank you for your comment. We have corrected terminologies as recommended.

  1. Give more extensive information on the limits of the research, including any possible confounding variables or sources of bias.

Response: Thank you for your comment. We have added more detailed relevant limitations.

  1. Consider incorporating a discussion section that gives a more in-depth examination and interpretation of the data.

Response: we have revised the discussion according to your suggestion.

  1. Use more current references to corroborate the study's conclusions, since the references given in the paper are fairly old.

Response: Thank you for your comment. We have now added recent references.

  1. Consider adding a remark about the clinical significance of the study's results and how they may be applied in practice.

Response: Thank you. We have added such comments.

  1. Give a clear and succinct description of the major results and implications of the research at the beginning of the article, in order to offer readers with an initial comprehension of the study's relevance.

Response: Thank you. We have added such comment as recommended.

  1. Overall, your work has the potential to make a substantial addition to the field of cardiology, and I advise you to address the areas for improvement to enhance your paper. Thank you again for your submission, and I look forward to reviewing the corrected paper.

Response: Thank you

Comments on the Quality of English Language

In the language and writing style, I propose editing the document to increase the clarity and readability of the language. This might involve utilising simpler language patterns, minimising jargon and technical vocabulary where feasible, and maintaining uniform formatting and terminology across the work for example the term "Worse".

Response: Thank you for your comment. We have revised the language as recommended.

Reviewer 2 Report

Dear editor

I read wit interest the article entitled “Strain rate changes during stress echocardiography is the most accurate predictor of significant coronary artery disease in patients with previouslytreated Acute Coronary Syndrom”. I conrats the authors for their study. however, some major limitations are available. My comments are below:

1- as the authors indicated, small number of the patients is a major lack. They stated that “We could not do power calculation becauseof the lack of previous publications in similar group of patients.” Maybe a retrospective power analysis may be performed. What was the power of the study retrospectively, after the end of the study. ıf the power was more than 80%, it may be usefull

2- the authors stated that they included ACS patients. I think that ACS types have major differences. Which ACS patients were included? The physiopathology of the STEMI and NSTEMI are quite different. To be the all patients STEMI or NSTEMI may be more useful fort he study? what was the number of ACS types? This is a major limitation

Dear editor

I read wit interest the article entitled “Strain rate changes during stress echocardiography is the most accurate predictor of significant coronary artery disease in patients with previouslytreated Acute Coronary Syndrom”. I conrats the authors for their study. however, some major limitations are available. My comments are below:

1- as the authors indicated, small number of the patients is a major lack. They stated that “We could not do power calculation becauseof the lack of previous publications in similar group of patients.” Maybe a retrospective power analysis may be performed. What was the power of the study retrospectively, after the end of the study. ıf the power was more than 80%, it may be usefull

2- the authors stated that they included ACS patients. I think that ACS types have major differences. Which ACS patients were included? The physiopathology of the STEMI and NSTEMI are quite different. To be the all patients STEMI or NSTEMI may be more useful fort he study? what was the number of ACS types? This is a major limitation

Author Response

Reviewer #2

Top of Form

Comments and Suggestions for Authors

Dear editor

I read with interest the article entitled “Strain rate changes during stress echocardiography is the most accurate predictor of significant coronary artery disease in patients with previously treated Acute Coronary Syndrome”. I congrats the authors for their study. However, some major limitations are available.

My comments are below:

1.As the authors indicated, small number of the patients is a major lack. They stated that “We could not do power calculation because of the lack of previous publications in similar group of patients.” Maybe a retrospective power analysis may be performed. What was the power of the study retrospectively, after the end of the study. If the power was more than 80%, it may be useful.

Response: Thank you for your comments and great suggestions. We now calculated the retrospective of sample size and power and also, we added the previous calculation for number of predictors accordingly to sample size.

Sample size and power:

Two patients’ groups (Significant vs. Non-significant; example for LAD)

For PSS:

Null hypothesis Ho: m1 = m2

Alternative hypothesis Ha: m1 = m2

β% of type II error: 0.84 at 80% statistical power and 1.28 at 90% statistical power

             = 4.8;

4.8 at power 80% and 12.7 at power 90% for each group.

For LVEF

Null hypothesis Ho: m1 = m2

Alternative hypothesis Ha: m1 = m2

β% of type II error: 0.84 at 80% power and 1.28 at 90% statistical power

             = 7.53;

7.53 at power 80% and 9.97 at power 90% for each group.

Global SR

Null hypothesis Ho: m1 = m2

Alternative hypothesis Ha: m1 = m2

β% of type II error: 0.84 at 80% power and 1.28 at 90% statistical power

             = 7.84;

7.84 at power 80% and 10 at power 90% for each group.

Reference: Suresh KP, Chandrashekara S. Sample size estimation and power analysis for clinical research studies. J Hum Reprod Sci. 2012; 5: 7–13.

In addition, because of the small sample size, to have a desired statistical power level about 0.8, we have calculated a priori sample size for multiple regression. Anticipated effect size (f2): 0.35; Desired statistical power level: 0.8; Number of predictors = 3; Probability level: 0.05; Minimum required sample size = 36

  1. The authors stated that they included ACS patients. I think that ACS types have major differences. Which ACS patients were included? The physiopathology of the STEMI and NSTEMI are quite different. To be the all patients STEMI or NSTEMI may be more useful fort the study? what was the number of ACS types? This is a major limitation

Response: Thank you for your comments. Our patients had pervious ACS and the time elapsed since the last ACS was 6.10 ± 3.9 years as we previously stated. They were treated with either PCI or CABG and they were symptoms free until recent admission. This admission was due to new onset of symptoms that suggested ischemia although they had negative cardiac biomarkers and normal pro-PNB. So they were considered to have chronic coronary syndrome and unstable angina, none of them had STEMI or NSTEMI at this admission.  

Reviewer 3 Report

Dear authors, I would like to make some constructive recommendations for your work, which are listed below.

Introduction

1. In the body of the introduction, adjectives are used such as: "very accurate to predict"

It is desirable not to use the word “very”, it is suggested: accurate to predict

methods

2. The correct abbreviation for microgram “40 mic of dobutamine/kg/min” is not used

Change to: dobutamine 40µg/kg/min

3. The words “in situ” are used in italics

Change for: in situ

4. The authors describe that their study "was approved by the local ethics committee"

It is desirable that the approval opinion number be included

5. Significance (p˂0.0)

Although some software outputs p˂0.0 as a result, it is desirable to add one thousandth p˂0.001

Results

6. The authors mention that: “a modest correlation was found”

It is suggested to change for: a poor or non-significant correlation was found

Discussion

7. The discussion should not list the results again.

8. Several lines of the discussion talk about "our patients"

It is suggested to change to: "study patients"

9. The authors state that the differences in the accuracy of the tested methods are attributed to the anatomical differences of the myocardial fibers, however, there is no citation or discussion to support the statement.

Change to is suggested: It is likely that the anatomical differences of the myocardial fibers are the cause of the differences in the accuracy of the tested methods.

10 limitations of the study

The authors speak of “obvious limitations”

It is suggested to change wording

11. The authors mention that their study is underpowered, however they conclude that myocardial deformation in the form of strain rate is a powerful predictor of culprit coronary artery injury.

It is suggested to change the wording of the limitations of the study, since the authors cannot support their conclusions.

12. The authors narrate experiments that they would have liked to do, but could do.

It is suggested to omit the wishes of the authors.

Author Response

Reviewer  Bottom of Form

#3

Comments and Suggestions for Authors

Dear authors, I would like to make some constructive recommendations for your work, which are listed below.

Introduction

  1. In the body of the introduction, adjectives are used such as: "very accurate to predict" It is desirable not to use the word “very”, it is suggested: accurate to predict

Response: Thank you for your comment. We have deleted the word ’very’.

 methods

  1. The correct abbreviation for microgram “40 mic of dobutamine/kg/min” is not used

Change to: dobutamine 40µg/kg/min

Response: Thank you for your comment. We have corrected this in the revised manuscript 

  1. The words “in situ” are used in italics

Change for: in situ

Response: Thank you for your comment. We have corrected this in the revised manuscript

  1. The authors describe that their study "was approved by the local ethics committee"

It is desirable that the approval opinion number be included

Response: Thank you for your comment. We have added the approval number.

  1. Significance (p˂0.0)

Although some software outputs p˂0.0 as a result, it is desirable to add one thousandth p˂0.001

Response: Thank you for your comment. We have corrected this in the revised manuscript

Results

  1. The authors mention that: “a modest correlation was found”

It is suggested to change for: a poor or non-significant correlation was found

Response: Thank you for your comment. We have changed it to ‘non-significant’

Discussion

  1. The discussion should not list the results again.

Response: Thank you for your comment. We have corrected this in the revised manuscript

  1. Several lines of the discussion talk about "our patients"

It is suggested to change to: "study patients"

Response: Thank you for your comment. We have corrected this.

  1. The authors state that the differences in the accuracy of the tested methods are attributed to the anatomical differences of the myocardial fibers, however, there is no citation or discussion to support the statement.

Change to is suggested: It is likely that the anatomical differences of the myocardial fibers are the cause of the differences in the accuracy of the tested methods.

Response: Thank you for your comment. We have corrected this in the revised manuscript

10 limitations of the study

The authors speak of “obvious limitations”

It is suggested to change wording

Response: Thank you for your comment. We have corrected it.

  1. The authors mention that their study is underpowered, however they conclude that myocardial deformation in the form of strain rate is a powerful predictor of culprit coronary artery injury.

It is suggested to change the wording of the limitations of the study, since the authors cannot support their conclusions.

Response: Thank you for your comment. We meant by underpowered the small number of patients studied, and despit that limitation we found the SR is a powerful predictor of the culprit artery.

  1. The authors narrate experiments that they would have liked to do, but could do.

It is suggested to omit the wishes of the authors.

Response: Thank you for your comment. We have omitted this part in the revised manuscript

Reviewer 4 Report

xxx

1. Please fix the title first. For example "previously treated" is written as previouslytreated and "acute coronary syndrome" is written as Acute Coronary Syndrom. This is incorrect.

2. In all instances, abbreviations from the title should be mentioned and abbreviated. This includes PSS, SR, WMSI, etc.

3. Authors do not provide ethical disclosures or Ethical Committee approval numbers for their research. This should be explained and provided.

4. Authors should clearly delineate that these patients were patients with chronic coronary syndrome and not an acute coronary syndrome.

5. How many of these patients had CTO? How did you differentiate between fresh occlusions and old occlusions?

6. Pharmacotherapy of patients at baseline should be provided. This information is currently unavailable.

7. It is intriguing that LVEF in your population was at a mean of 36.2% which suggests that many of these patients had heart failure due to ischemic cardiomyopathy. Was heart failure handled appropriately in this cohort?

8. What would be the rationale for revascularization of patients with left ventricular dysfunction and ischemic cardiomyopathy through the eyes of the latest data (REVIVED study, etc.)?

9. Why didn't you use troponin and other cardioselective enzymes that might help you differentiate between acute vs. chronic myocardial injury? How can you be sure how old were these lesions?

10. I would be interested to know NT-proBNP in these patients.

11. 

English is generally fine, although there are technical typos throughout the manuscript. I suggest authors to recheck their manuscript thoroughly.

Author Response

Reviewer #4

Comments and Suggestions for Authors

1.Please fix the title first. For example "previously treated" is written as previouslytreated and "acute coronary syndrome" is written as Acute Coronary Syndrom. This is incorrect.

Response: Thank you for your comment. We have corrected the title in the revised manuscript

  1. In all instances, abbreviations from the title should be mentioned and abbreviated. This includes PSS, SR, WMSI, etc.

Response: Thank you for your comment. We have implemented all abbreviations.

3.Authors do not provide ethical disclosures or Ethical Committee approval numbers for their research. This should be explained and provided.

Response: Thank you for your comment. We have added the ethics approval number.

4.Authors should clearly delineate that these patients were patients with chronic coronary syndrome and not an acute coronary syndrome.

Response: Thank you for your comment. We have now added more clarification for the inclusion and exclusion criteria. This group of patients had pervious acute coronary syndrome and the time elapsed since the last ACS was 6.10 ± 3.9 years as we stated in the paper. They were treated with either PCI or CABG and they were symptom free until recent admission. This admission was due to new onset of symptoms that suggests ischemia although they had negative cardiac biomarkers and none had STEMI or NSTEMI on the recent admission. 

  1. How many of these patients had CTO? How did you differentiate between fresh occlusions and old occlusions?

Response: Thank you. The culprit lesion was defined as more than 70% arterial occlusion.  We considered patients with 100% occlusion as the culprit artery. Regarding chronicity of the occlusion we did not evaluate it, but as we stated in the paper there were 17 patients with 100% LAD occlusion, 8 patients with LCx occlusion and 8 patients with RCA occlusion.

  1. Pharmacotherapy of patients at baseline should be provided. This information is currently unavailable.

Response: Thank you for your comment. All patients were on the usual anti ischaemic medications (ASA, statins, Beta Blockers, ACE-I/ARBS and furosemide. We Have now added medications.

7.It is intriguing that LVEF in your population was at a mean of 36.2% which suggests that many of these patients had heart failure due to ischemic cardiomyopathy. Was heart failure handled appropriately in this cohort?

Response: Thank you. The studied patients presented with symptoms of either dyspnea, chest pain or both. We excluded other causes of dyspnea as heart failure, severe anemia, chest infection particularly as this study was during the Covid-19, patients with overt signs of heart failure were excluded from the study.

8.What would be the rationale for revascularization of patients with left ventricular dysfunction and ischemic cardiomyopathy through the eyes of the latest data (REVIVED study, etc.)?

Response: Thank you for your comment. The REVIVED-BCIS2 trial failed to show that multivessel PCI improved event-free survival and LVEF among patients with severe ischemic cardiomyopathy. Patients had PCI for symptoms.

9.Why didn't you use troponin and other cardio selective enzymes that might help you differentiate between acute vs. chronic myocardial injury? How can you be sure how old were these lesions?

Response: Thank you. The serial Troponin and CK-MB tests were done for all to differentiate patients with STEMI/NSTEMI from those recruited in this study. We have now added this to the text.

10.I would be interested to know NT-pro BNP in these patients.

Response: Thank you. The recruited patients had normal pro-BNP level.

Comments on the Quality of English Language

English is generally fine, although there are technical typos throughout the manuscript. I suggest authors recheck their manuscript thoroughly.

Response: Thank you for your comment. We have now checked it all again.

Reviewer 5 Report

The manuscript is well written and very interesting.

I suggest the Authors to correct all the refusals within all the body of the manuscript. For example in the Introduction section: “after/myocardial infarction; prior to/coronary revascularization ecc”.

I respectfully disagree with the Authors’ choice to perform dobutamine stress echocardiography only, for evaluating patients with history of CAD. The Authors did not consider exercise stress echocardiography. The latter should be preferred to dobutamine stress echocardiography in the clinical practice for evaluating patients with history of CAD, especially if they have not motor deficits, as indicated by the international guidelines. The reason for this choice should be explained in the Limitations section.

Moreover, in the paragraph “Clinical implications” the Authors could add the following sentence: “Despite the limitations of speckle tracking echocardiography, such as the temporal stability of tracking patterns, the intervendor variability, the dependency of deformation on chamber geometry, dyssynchrony and segment interactions, it should be implemented in the clinical practice for evaluating symptomatic patients with history of CAD during stress echocardiography”. Please cite the following references: “Voigt JU, Pedrizzetti G, Lysyansky P, Marwick TH, Houle H, Baumann R, Pedri S, Ito Y, Abe Y, Metz S, Song JH, Hamilton J, Sengupta PP, Kolias TJ, d'Hooge J, Aurigemma GP, Thomas JD, Badano LP. Definitions for a common standard for 2D speckle tracking echocardiography: consensus document of the EACVI/ASE/Industry Task Force to standardize deformation imaging. J Am Soc Echocardiogr. 2015 Feb;28(2):183-93. doi: 10.1016/j.echo.2014.11.003. PMID: 25623220. Mirea O, Pagourelias ED, Duchenne J, Bogaert J, Thomas JD, Badano LP, Voigt JU; EACVI-ASE-Industry Standardization Task Force. Intervendor Differences in the Accuracy of Detecting Regional Functional Abnormalities: A Report From the EACVI-ASE Strain Standardization Task Force. JACC Cardiovasc Imaging. 2018 Jan;11(1):25-34. doi: 10.1016/j.jcmg.2017.02.014. Epub 2017 May 17. PMID: 28528162. Sonaglioni A, Nicolosi GL, Rigamonti E, Lombardo M, La Sala L. Molecular Approaches and Echocardiographic Deformation Imaging in Detecting Myocardial Fibrosis. Int J Mol Sci. 2022 Sep 19;23(18):10944. doi: 10.3390/ijms231810944. PMID: 36142856; PMCID: PMC9501415)”.

I suggest the Authors to correct all the refusals within all the body of the manuscript. For example in the Introduction section: “after/myocardial infarction; prior to/coronary revascularization ecc”.

Author Response

Reviewer #5

Comments and Suggestions for Authors

The manuscript is well written and very interesting.

I suggest the Authors to correct all the refusals within all the body of the manuscript. For example in the Introduction section: “after/myocardial infarction; prior to/coronary revascularization ecc”.

I respectfully disagree with the Authors’ choice to perform dobutamine stress echocardiography only, for evaluating patients with history of CAD. The Authors did not consider exercise stress echocardiography. The latter should be preferred to dobutamine stress echocardiography in the clinical practice for evaluating patients with history of CAD, especially if they have not motor deficits, as indicated by the international guidelines. The reason for this choice should be explained in the Limitations section.

Response: Thank you for your comment. The mean age of the studied population was 60+-10 year and many of them are overweight and having other comorbidities that made them unable to exercise well, so we choose DSE, we added this to the limitations

Moreover, in the paragraph “Clinical implications” the Authors could add the following sentence: “Despite the limitations of speckle tracking echocardiography, such as the temporal stability of tracking patterns, the intervendor variability, the dependency of deformation on chamber geometry, dyssynchrony and segment interactions, it should be implemented in the clinical practice for evaluating symptomatic patients with history of CAD during stress echocardiography”. Please cite the following references: “Voigt JU, Pedrizzetti G, Lysyansky P, Marwick TH, Houle H, Baumann R, Pedri S, Ito Y, Abe Y, Metz S, Song JH, Hamilton J, Sengupta PP, Kolias TJ, d'Hooge J, Aurigemma GP, Thomas JD, Badano LP. Definitions for a common standard for 2D speckle tracking echocardiography: consensus document of the EACVI/ASE/Industry Task Force to standardize deformation imaging. J Am Soc Echocardiogr. 2015 Feb;28(2):183-93. doi: 10.1016/j.echo.2014.11.003. PMID: 25623220. Mirea O, Pagourelias ED, Duchenne J, Bogaert J, Thomas JD, Badano LP, Voigt JU; EACVI-ASE-Industry Standardization Task Force. Intervendor Differences in the Accuracy of Detecting Regional Functional Abnormalities: A Report From the EACVI-ASE Strain Standardization Task Force. JACC Cardiovasc Imaging. 2018 Jan;11(1):25-34. doi: 10.1016/j.jcmg.2017.02.014. Epub 2017 May 17. PMID: 28528162. Sonaglioni A, Nicolosi GL, Rigamonti E, Lombardo M, La Sala L. Molecular Approaches and Echocardiographic Deformation Imaging in Detecting Myocardial Fibrosis. Int J Mol Sci. 2022 Sep 19;23(18):10944. doi: 10.3390/ijms231810944. PMID: 36142856; PMCID: PMC9501415)”.

Response: Thank you for your comment. We have added the references

Comments on the Quality of English Language

I suggest the Authors to correct all the refusals within all the body of the manuscript. For example in the Introduction section: “after/myocardial infarction; prior to/coronary revascularization ecc”.

Response: Thank you for your comment. We have corrected this in the revised manuscript

Round 2

Reviewer 2 Report

the authors responded in a satisfactory way. 

the authors responded in a satisfactory way. 

Reviewer 3 Report

Estimados autores del artículo diagnostico-2329529, los felicito por su trabajo

Reviewer 4 Report

Thank you for addressing my comments. No further comments.

In this current for, article might be suitable for publication.